# Phosphorylation of tyrosine 90 in SH3 domain is a new regulatory switch controlling Src kinase

**Lenka Koudelková[1,2], Markéta Pelantová[1], Zuzana Brůhová[1], Martin Sztacho[1], Vojtěch Pavlík[1], Dalibor Pánek[3], Jakub Gemperle[1], Pavel Talacko[4], Jan Brábek[1], Daniel Rösel[1]***

[1]Department of Cell Biology, BIOCEV, Faculty of Science, Charles University, Vestec, Czech Republic; [2]Institute of Organic Chemistry and Biochemistry, Czech Academy of Sciences, Prague, Czech Republic; [3]Imaging Methods Core Facility at BIOCEV, Faculty of Science, Charles University, Vestec, Czech Republic; [4]Proteomics Core Facility at BIOCEV, Faculty of Science, Charles University, Vestec, Czech Republic

*For correspondence:
rosel@natur.cuni.cz

**Competing interest:** The authors declare that no competing interests exist.

**Abstract** The activation of Src kinase in cells is strictly controlled by intramolecular inhibitory interactions mediated by SH3 and SH2 domains. They impose structural constraints on the kinase domain holding it in a catalytically non-permissive state. The transition between inactive and active conformation is known to be largely regulated by the phosphorylation state of key tyrosines 416 and 527. Here, we identified that phosphorylation of tyrosine 90 reduces binding affinity of the SH3 domain to its interacting partners, opens the Src structure, and renders Src catalytically active. This is accompanied by an increased affinity to the plasma membrane, decreased membrane motility, and slower diffusion from focal adhesions. Phosphorylation of tyrosine 90 controlling SH3-mediated intramolecular inhibitory interaction, analogical to tyrosine 527 regulating SH2-C-terminus bond, enables SH3 and SH2 domains to serve as cooperative but independent regulatory elements. This mechanism allows Src to adopt several distinct conformations of varying catalytic activities and interacting properties, enabling it to operate not as a simple switch but as a tunable regulator functioning as a signalling hub in a variety of cellular processes.

## Editor's evaluation

This manuscript reports that a phosphomimetic mutation of a previously unstudied phosphorylation site in the Src kinase SH3 domain (Y90) elevates Src kinase activity via loosening the conformation of the Src catalytic domain. This work establishes a solid foundation for the further analysis of how and under what circumstances this modification impacts Src kinase functions in myriad biological contexts. This paper will be of interest to those studying protein kinase domain dynamicity and structure, and also those studying the Src kinase family in cell signaling.

## Introduction

Src kinase is a founding member of the Src family nonreceptor tyrosine kinases (SFKs) that are crucial for a large variety of signalling and metabolic pathways such as cell differentiation, proliferation, survival, motility, and mechanosignalling (*Guarino, 2010*; *Koudelková et al., 2021*; *Thomas and Brugge, 1997*; *Yeatman, 2004*). The significance of these processes in cellular homeostasis therefore implies a requirement for precise and tight regulation of the kinase. Aberrant Src activation leads to

cellular transformation and has been found in several types of cancers including gastric, lung, pancreatic, ovarian, or breast neoplasms (*Frame, 2002*; *Irby and Yeatman, 2000*).

Src and other SFKs share common multidomain architecture consisting of a myristoylated SH4 domain at the very N-terminus followed by a unique domain (UD), regulatory SH3 and SH2 domains, a CD linker, a catalytic domain, and a C-terminal tail containing the regulatory residue tyrosine 527 (Y527; chicken c-Src numbering will be used throughout the paper). The activity of the Src kinase is regulated in response to numerous cellular signals through allosteric conformational transitions. In the inactive state, Src adopts a closed conformation with the SH3 domain interacting with the CD linker, and the SH2 domain bound to the C-terminus phosphorylated on Y527. These autoinhibitory interactions pack both regulatory domains against the kinase domain at a site opposite to the catalytic cleft. In this position, SH3 and SH2 domains form contacts with lobes of the kinase domain, working as a clamp which locks the kinase domain in a catalytically non-permissive state characterised by the activation loop folded inside the catalytic cleft and the αC-helix of the N-lobe rotated away from the active site (*Boggon and Eck, 2004*; *Xu et al., 1999*; *Xu et al., 1997*; *Young et al., 2001*). The shift to an open active conformation is accompanied by a release of intramolecular inhibitory interactions, which enables the kinase domain to adopt a catalytically favourable state with the unfolded activation loop accessible for phosphorylation of a regulatory tyrosine 416 (Y416) and the αC-helix rotated towards the active site (*Cowan-Jacob et al., 2005*). Kinase activity and engagement of regulatory domains into autoinhibitory interactions are controlled by the presence of ligands for SH3 and SH2 domains and by the phosphorylation status of two regulatory tyrosines, Y527 and Y416, with opposing effects. Phosphorylated Y527 in the C-terminal tail associates with the SH2 domain, resulting in an inhibition of kinase activity. The phosphorylation of Y416 in the activation loop stabilises the active conformation of the kinase domain and is necessary for full catalytic activity (*Roskoski, 2005*).

The role and mechanism of the inhibitory interaction between the SH2 domain and phosphorylated Y527 is well established (*Cowan-Jacob et al., 2005*; *Roskoski, 2005*). The regulation of the SH3 domain engagement, however, appears to be more complex and less understood. Besides binding a proline motif within the CD linker, the SH3 domain forms another set of inhibitory contacts with the N-lobe of the kinase domain via the RT loop and the nSrc loop (*Brábek et al., 2002*). Additionally, together with the SH2 domain, the SH3 domain holds the CD linker in a position which stabilises the inactive conformation of the catalytic domain (*Fajer et al., 2017*). Furthermore, the SH3 domain associates with membranes and provides structural support for intrinsically disordered SH4 and UD. Therefore, the SH3 domain seems to operate as a hub interconnecting signals from the flexible membrane-bound N-terminal part of Src together with information from the structured units of the kinase (*Maffei et al., 2015*; *Pérez et al., 2013*). Despite the multitude of interactions and signalling events known to be mediated by Src domains, the full complexity of the Src kinase regulatory network, including the hierarchy and autonomy of individual structural elements or their exact effect on Src conformation, remains to be understood.

In this study, we propose a novel mechanism of Src kinase regulation via phosphorylation of the SH3 domain on the tyrosine 90 (Y90). Using phosphomimicking or nonphosphorylatable substitutions of Y90, we showed that phosphorylation of Y90 increases Src kinase activity, reduces its affinity for SH3 domain ligands, induces the open conformation of the kinase, and slows down its molecular mobility within the cytoplasmic membrane. Furthermore, deregulating Y90 phosphorylation dynamics with a phosphomimicking mutation induces cellular transformation and increases the invasive potential of cells.

## Results

### Substitutions mimicking the Y90 phosphorylation state affect Src kinase activity

In a proteomic study which compared phosphoproteomes of mouse embryonic fibroblasts (MEFs) and Src-transformed MEFs (*Luo et al., 2008*), Y90 was identified as a novel site of tyrosine phosphorylation within the Src kinase. Y90 is localised on the binding surface of the Src SH3 domain and participates in the formation of the hydrophobic pocket involved in ligand binding, including the CD linker of Src. The interaction between the SH3 domain and the CD linker is one of the regulatory switches participating in the maintenance of the inactivated state of the kinase (*Boggon and Eck, 2004*; *Xu*

*et al., 1997*). Therefore, changes within the binding surface of the SH3 domain, such as phosphorylation of one of the amino acid residues, might affect Src regulation.

In order to analyse the role of Y90 phosphorylation on Src kinase activity, we prepared mutational variants of Src with either the phosphomimicking substitution of Y90 to glutamate (90E) or the nonphosphorylatable substitution of Y90 to phenylalanine (90F). These mutants, together with wild type (WT) Src (c-Src) and constitutively active Src (Src527F), were stably expressed in SYF cells, a lineage of MEFs with deleted genes for Src family kinases Src, Yes, and Fyn (*Klinghoffer et al., 1999*).

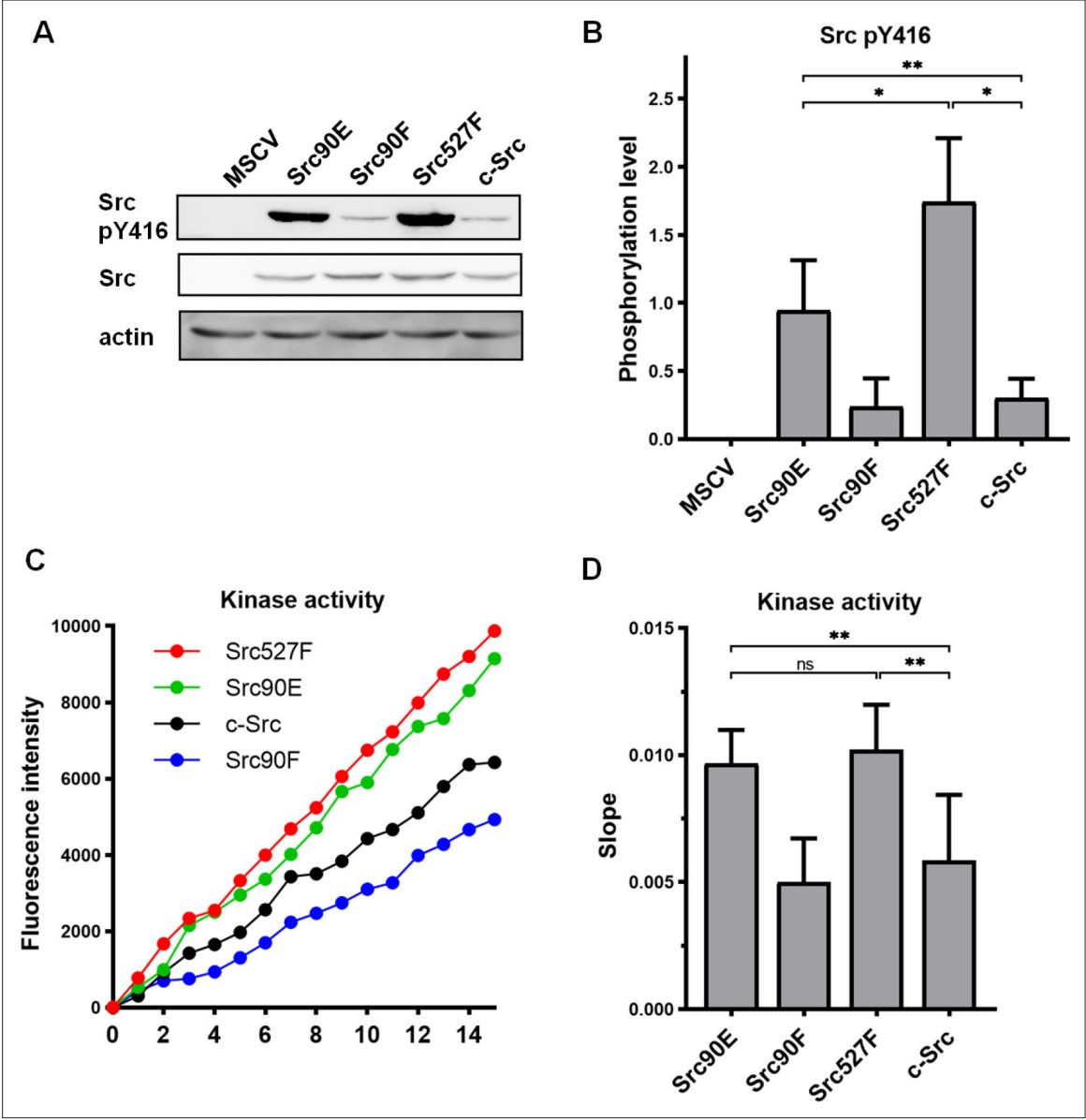

**Figure 1.** Substitutions mimicking the Y90 phosphorylation state affect Src kinase activity. (**A, B**) Lysates from SYF fibroblasts stably expressing the Src variants were analysed by immunodetection on western blots using antibodies against Src pY416, total Src, and actin as a loading control. After densitometric analysis of the blots, ratio between Y416-phosphorylated Src and total Src was calculated. MSCV represents SYF cells bearing empty pMSCV-EGFP vector. (**C, D**) Kinase assay (Omnia Y Peptide 2 Kit) was performed from lysates of SYF cells expressing the Src variants. Kinase activity was measured as an increase of fluorescence depicted in **C** (showing representative results from one experiment). The data are shown as means with standard deviation out of minimum four independent experiments. Statistical significance was calculated by one-way ANOVA.

The online version of this article includes the following source data for figure 1:

**Source data 1.** Unedited immunoblots and source datasets for graphs in *Figure 1*.

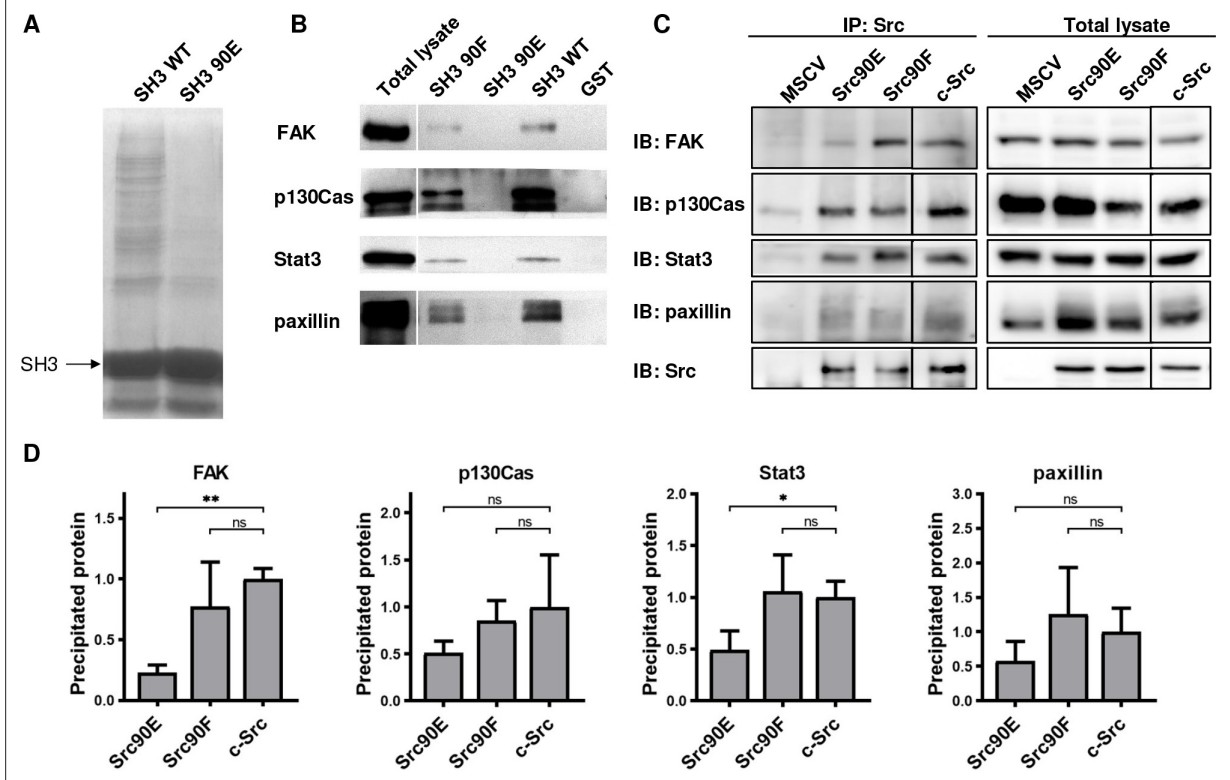

**Figure 2.** Phosphomimicking mutation 90E reduces the SH3 domain ligand binding capacity. The variants of Src SH3 domains (90F, 90E, wild type [WT]) were expressed as recombinant GST-SH3 fusion constructs in bacteria, isolated and used for pull-down from HeLa lysates. (**A**) Precipitated samples were run on SDS-PAGE electrophoresis and stained with Coomassie Brilliant Blue. Loading of GST-SH3 constructs was equal and is visible as a large band with molecular weight of around 33 kDa. (**B**) Western blotting of the precipitates was followed by immunodetection by antibodies against FAK, p130Cas, Stat3, or paxillin. (**C, D**) Immunoprecipitation of the full-length Src variants from SYF lineages followed by immunodetection of selected SH3-binding proteins in precipitates and total cell lysates. MSCV indicates SYF cell expressing empty pMSCV-EGFP construct, which serve as a control. Quantification was performed from at least three independent experiments and represents a normalised ratio between immunoprecipitated protein and total protein. Statistical significance was calculated by one-way ANOVA.

The online version of this article includes the following source data for figure 2:

**Source data 1.** Unedited immunoblots and source datasets for graph in *Figure 2*.

Initially, we tested the effect of the phosphomimicking substitution 90E on the phosphorylation of the Src activation loop tyrosine 416, which is known to correlate with Src kinase activity (*Boerner et al., 1996*). We found that the 90E substitution results in a significant increase of Y416 phosphorylation, suggesting that phosphorylation of Y90 increases Y416 autophosphorylation of Src (*Figure 1A and B*).

Next, we analysed the direct effect of 90E and 90F substitutions on the kinase activity of Src. Src activity was measured in lysates prepared from SYF cells stably expressing the Src variants. Consistent with the analysis of Y416 autophosphorylation, we found the kinase activity of the Src90E variant to be significantly increased relative to that of c-Src. The nonphosphorylatable variant Src90F consistently displayed slightly lower, albeit not statistically significant, kinase activity than WT c-Src (*Figure 1C and D*).

Taken together, these results suggest that Y90 within the Src SH3 domain is involved in the regulation of Src Y416 autophosphorylation and kinase activity.

## Phosphomimicking mutation 90E reduces the SH3 domain ligand binding capacity

Elevated kinase activity of Src90E suggests that there may be impaired association of the CD linker to the SH3 domain. To further examine the effect of Y90 substitutions on the binding properties of the SH3 domain, we performed pull-down assays with GST-fused Src SH3 domain variants (WT, 90E, 90F).

The number of proteins precipitated via GST-SH3-90E was greatly reduced compared to the GST-SH3-WT sample (*Figure 2A*). Moreover, the phosphomimicking substitution 90E almost completely abolished the binding of FAK, p130Cas, Stat3, and paxillin, the key interacting proteins of Src SH3 domain. Conversely, the 90F mutation had a minor or no effect on ligand interaction, except for FAK binding, which was partially abrogated (*Figure 2B*). These results suggest that phosphorylation of Y90 inhibits Src ability to interact with its signalling partners.

To assess the impact of the altered affinity of the SH3 domain on the binding potential of the full-length kinase, we immunoprecipitated Src variants from SYF lineages and detected the amount of SH3-interacting proteins. We observed substantially decreased association of the phosphomimicking mutant Src90E to FAK and Stat3 when compared to c-Src. Affinity of full-length Src90E to paxillin and p130Cas was only slightly weakened, potentially due to significant contribution of Src SH2 domain to the interaction between these proteins. Src with the nonphosphorylatable substitution 90F exhibited similar binding to the selected SH3-interacting proteins as c-Src (*Figure 2C and D*).

## Phosphorylation of Y90 opens the Src structure

Elevated kinase activity and reduced ligand binding of the phosphomimicking mutant Src90E indicate that Y90 phosphorylation could also decrease the affinity of the SH3 domain for the CD linker, thus potentially causing a loosening of the compact conformation maintained by inhibited Src. To evaluate the effects of Y90 substitutions on Src structure, we assessed the compactness of the Src variants by employing an SrcFRET biosensor (*Koudelková et al., 2019*). The SrcFRET biosensor allows for reading of Src conformational changes by monitoring an intramolecular FRET between a donor mCFP and acceptor mCit fluorophores inserted into the SH2 domain and at the very C-terminus of the kinase, respectively. In the inactive compact conformation, the fluorophores are in close proximity, which results in high values of FRET. Upon activation, the Src structure loosens, causing separation of the fluorophores and a decrease of FRET.

The sequence of the original SrcFRET sensor was modified by introducing the Y90 phospho-mimicking and nonphosphorylatable mutations or the 527F activating mutation. We expressed the constructs in the U2OS cell line and measured FRET in lysates using a ratiometric approach. The nonphosphorylatable 90F version exerted higher FRET than the WT sensor, indicating a more compact conformation. Conversely, the phosphomimicking mutation 90E led to a decrease in FRET to an intermediate level between WT SrcFRET and 527F-mutated SrcFRET (SrcFRET527F), therefore opening the Src structure but to a lower extent compared to SrcFRET527F (*Figure 3A and B*).

## Phosphorylation of Y90 occurs by autophosphorylation

To determine the level of Src phosphorylation on Y90, we performed quantitative mass spectrometry (MS) analysis with immunoprecipitated WT SrcFRET and SrcFRET527F. To obtain information on the absolute level of phosphorylated molecules, we used stable isotope-labelled phospho- and nonphospho-peptides of known concentrations that served as standards allowing calibration of measurements. We were able to detect phosphorylation of Y416 in 22% of WT Src molecules, which further increased to 57% in case of constitutively active form of Src. Phosphorylation of Y90 was present only in 1% of WT molecules but became five times more abundant upon kinase activation, as demonstrated with the SrcFRET527F variant (*Figure 3C*).

Based on bioinformatic predictions, Y90 is likely to be phosphorylated by Src family kinases or Src itself (*Tatárová et al., 2012*). To support this notion, we took advantage of an established panel of Src activatory mutations, which we evaluated for their effect on structural compactness and catalytic activity by introducing them in the SrcFRET sensor (*Koudelková et al., 2019*). For subsequent analyses, we selected three SrcFRET variants exhibiting distinct kinase activities: the original sensor with unmutated Src, representing a molecule with the lowest activity, SrcFRET527F with the intermediately high activity and SrcFRET-CA (Glu381Gly, *Bjorge et al., 1995*) possessing the highest activity. Chosen variants were analysed for Y90 phosphorylation levels using quantitative MS. Y90 phosphorylation was detected in all variants, with 5- and 12-fold higher incidence compared to WT in activated constructs SrcFRET527F and SrcFRET-CA, respectively (*Figure 3D*). The revealed positive correlation between Src catalytic activity and Y90 phosphorylation levels suggesting that Y90 is phosphorylated through autophosphorylation mechanisms. To further address the question of Y90 autophosphorylation, we expressed a kinase dead version of the Src527F biosensor with a K295M substitution (SrcFRET527F

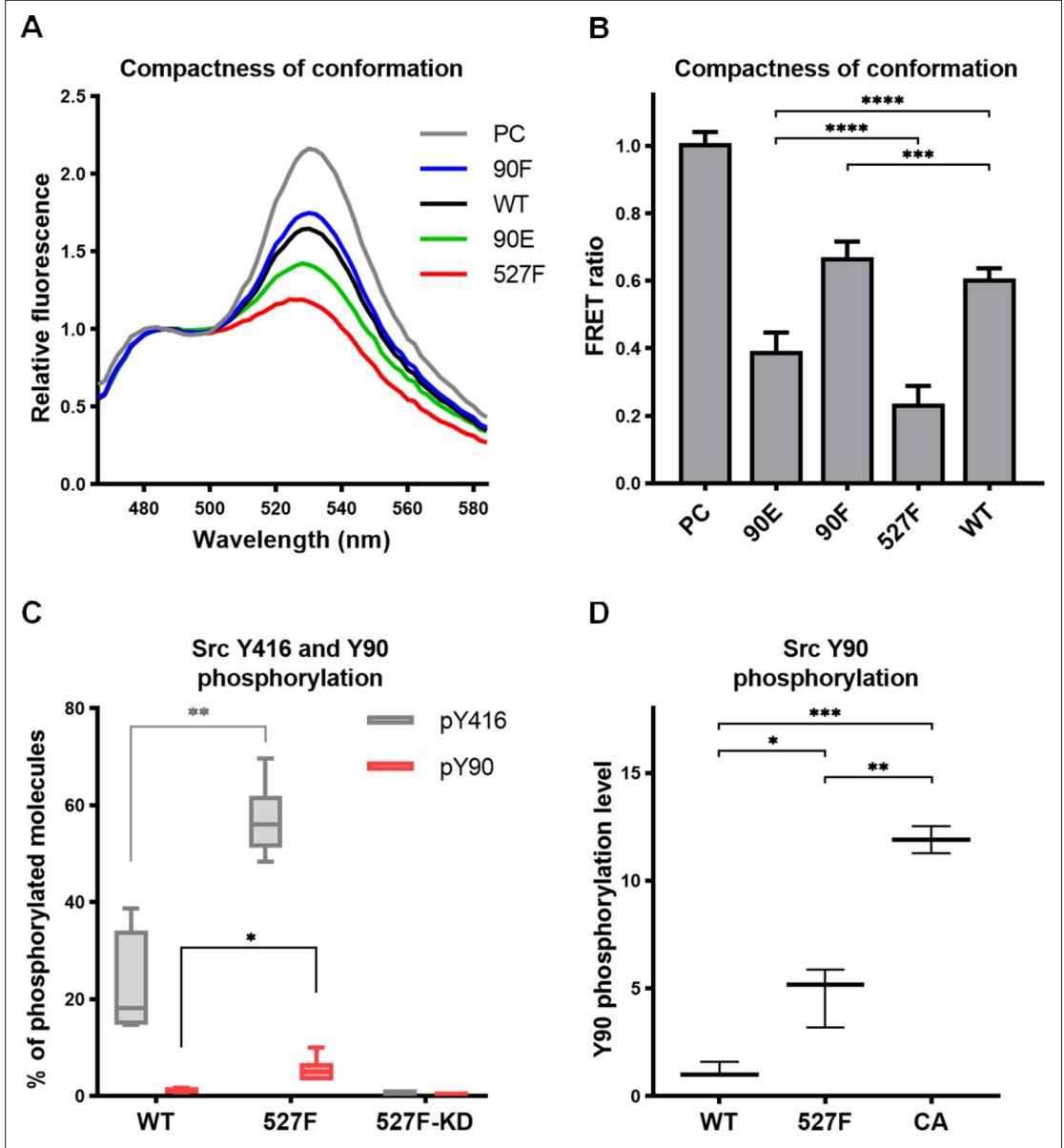

**Figure 3.** Phosphorylation of Y90 represents an autoactivatory mechanism that opens the Src structure. (**A, B**) Analysis of effects mediated by the Y90 substitutions on the Src kinase structure using SrcFRET sensor. SrcFRET sensor variants with Y90 and/or Y527 mutations were expressed in U2OS cells. Fluorescence emission spectra were recorded in native cell lysates. (**A**) Representative emission spectra normalised to emission maximum of CFP. PC indicates positive control for FRET (construct where CFP and mCit are connected by a short linker). (**B**) Bar graph shows ratio of normalised mCit (525 nm) and CFP (486 nm) emission. The data are shown as means with standard deviation out of minimum four independent experiments. Statistical significance was calculated by one-way ANOVA. (**C, D**) Quantitative mass spectrometry (MS) analysis of Src phosphorylation. SrcFRET constructs were expressed in U2OS cells, immunoprecipitated and analysed on MS. (**C**) Absolute values of Y416 and Y90 phosphorylation were determined using internal peptide standards and depicted on the plot as percentage of phosphorylated tyrosines from the total pool of given tyrosine. (**D**) Normalised ratio between phopho-peptide and non-modified base peptide was plotted. The data are shown as means with standard deviation from three independent experiments. Statistical significance was calculated by one-way ANOVA.

The online version of this article includes the following source data and figure supplement(s) for figure 3:

**Source data 1.** Source datasets for graphs in *Figure 3*.

**Figure supplement 1.** Activated Y527F Src carrying a kinase-dead mutation retains an open conformation.

KD) and analysed the phosphorylation levels of Y90 and Y416 using quantitative MS. Despite the inactivating mutation, SrcFRET527F KD retained an open conformation (*Figure 3—figure supplement 1A and B*). However, phosphorylation of both tyrosines in the kinase dead variant was negligible even though both endogenous Src and other SFKs are present in the U2OS cells we used for the experiment (*Figure 3C*). Taken together, these results indicate that the phosphorylation of Y90 is dependent on the intrinsic kinase activity of Src and is therefore very likely to be autophosphorylation.

## Y90 phosphorylation state does not affect Src localisation but alters its mobility within the plasma membrane

Introduction of the Y90 mutations to the SrcFRET biosensor allowed us to analyse the effect of these substitutions on Src localisation. Neither mutation, 90E or 90F, led to any apparent changes in Src cellular localisation. SrcFRET90E and SrcFRET90F were uniformly distributed in the cell, with enrichment in membrane ruffles, the same as SrcFRET. Constitutively active Src, which carries the 527F mutation, strongly localised to focal adhesions (FAs). In contrast to SrcFRET527F, phosphomimicking SrcFRET90E, despite its increased activation and open conformation, did not exhibit any apparent enrichment in adhesion sites (*Figure 4A*).

Src localisation to plasma membrane is secured by a myristic acid anchor and by electrostatic interactions between the lipid layer and the N-terminal part of the molecule involving the SH4, unique, and SH3 domains (*Pellman et al., 1985*; *Pérez et al., 2013*). Additionally, the Src SH3 domain acts as a scaffold providing structural support to this intrinsically disordered segment of the kinase. The binding of a ligand to the SH3 domain was reported to abrogate its interaction with the UD and lipids, leading to elevated conformational flexibility of the UD, reduced contacts with the membrane, and potentially to increased dynamics within the lipid layer (*Machiyama et al., 2015*; *Maffei et al., 2015*; *Pérez et al., 2013*). To determine the effect of the Y90 phosphorylation state on the mobility of membrane-attached Src, we performed imaging fluorescence correlation spectroscopy (ImFCS) measurements combined with TIRF microscopy. This setup allowed us to assess dynamics of highly motile Src molecules by measuring temporally and spatially resolved fluctuations of fluorescence signal emitted by fluorophores attached to our Src variants. Using TIRF microscopy ensured that collected signal originated from the plasma membrane in close proximity to the coverslip. Correlation curves calculated from obtained data were used to determine diffusion coefficient of Src molecules.

All tested Src variants exhibited decreased mobility in FA sites compared to membrane regions outside of FAs (*Figure 4B*; note the shift of 'FA' correlation curves in *Figure 4—figure supplement 1A and B* to the right). This indicates transient stabilisation of the kinase within FAs. The phosphomimicking mutation 90E significantly reduced Src motion within the lipid layer compared to c-Src, both outside and inside FAs, with more prominent decrease detected in FA areas. Diffusion coefficients of Src with the nonphosphorylatable mutation 90F were similar to those of c-Src. 527F-mutated Src showed comparable motility to Src90E outside FAs and marked retention inside FAs, corresponding to its pronounced accumulation in adhesion sites.

Lower dynamics of the phosphomimicking mutant Src90E in the plasma membrane suggest that the decreased affinity of the SH3 domain towards its ligands caused by the phosphorylation of Y90 leads to slower motility of Y90-phosphorylated Src within the cytoplasmic membrane and prolonged residence in FA sites.

## Src90E has transforming potential

Deregulated and therefore more active forms of the Src kinase are capable of inducing cellular transformation. Cells transformed with activated Src typically exhibit a loss of contact inhibition and gain the ability to grow independently of attachment (*Frame, 2002*; *Thomas and Brugge, 1997*). As shown above, Src90E possesses elevated kinase activity and, at the same time, reduced SH3 domain ligand binding. To analyse whether the elevated Src kinase activity of the 90E variant is able to induce transformation even while the SH3 domain has impaired ability to bind ligands, we performed a soft agar assay with the Src variants. Normal fibroblasts are not capable of growing in such an environment and die by anoikis, while transformed cells survive and proliferate (*Cifone, 1982*; *Macpherson and Montagnier, 1964*).

As expected, the highest number of colonies was induced in SYF fibroblast expressing the constitutively active Src527F. SYF cells bearing just the empty vector (SYF-MSCV) were not able to form

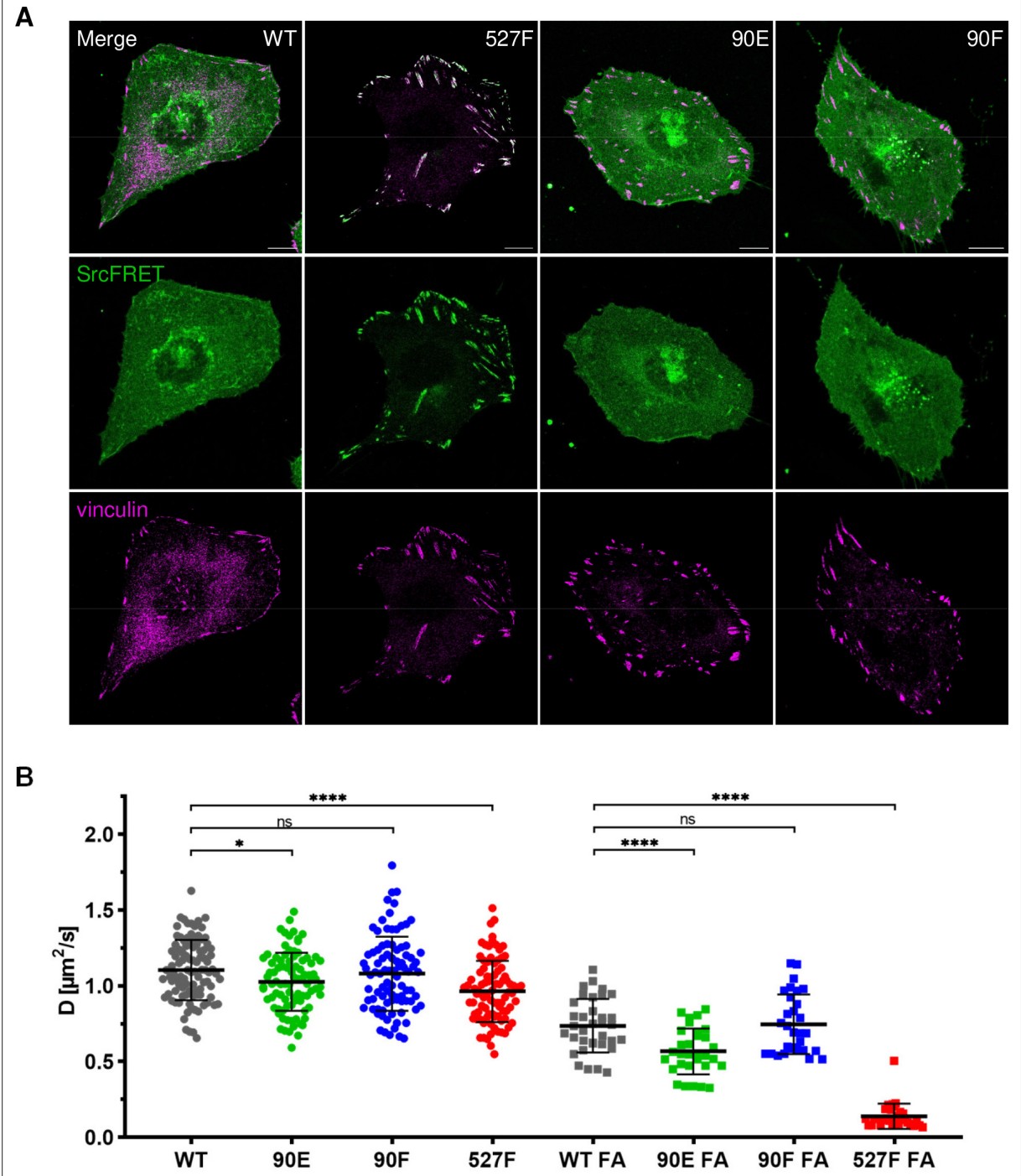

**Figure 4.** Y90 phosphorylation state does not affect Src localisation but alters its mobility within cytoplasmic membrane. (**A**) U2OS cells transiently expressing the FRET constructs were fixed, stained for vinculin (focal adhesions marker, magenta) and imaged by confocal microscopy. SrcFRET fluorescence (mCit) is depicted in green. Scale bar: 10 μm. (**B**) Diffusion coefficients of membrane associated SrcFRET variants outside and inside focal adhesions. U2OS cells were cotransfected with SrcFRET constructs and mCherry-vinculin. Mobility of the SrcFRET variants within the membrane was measured by imaging fluorescence correlation spectroscopy (ImFCS) combined with TIRF microscopy. Membrane regions within focal adhesions were identified by the presence of mCherry-vinculin. Data from three independent experiments are shown as a scatter plot with the mean represented by the middle horizontal line. Statistical significance was calculated by one-way ANOVA.

The online version of this article includes the following source data and figure supplement(s) for figure 4:

**Source data 1.** Source dataset for graph in *Figure 4B*.

**Figure supplement 1.** Y90 phosphorylation state does not affect Src localisation but alters its mobility within cytoplasmic membrane.

any colonies. In the majority of experiments no colonies were formed by SYF-Src90F cells either. SYF fibroblasts with the phosphomimicking mutant Src90E were able to grow independently of attachment profoundly substantially better than SYF cells with WT c-Src, although less effectively when compared to SYF-Src527F. These results indicate that Src90E has a transforming potential (*Figure 5A and B*).

In control experiments, we analysed the proliferation rates in 2D. Expression of all Src variants leads to increased proliferation of SYF fibroblasts. Although we did not observe any significant changes among the variants themselves, SYF-Src90E and SYF-Src527F proliferated slightly faster compared to SYF-Src90F and SYF-c-Src cells (*Figure 5C*).

Transformed cellular phenotype is a consequence of aberrant activation of mitogenic signalling pathways, which are normally under the control of Src. Crucial contribution to transformation mediated by Src is ascribed to Ras/MAPK, PI3K/Akt, and Stat3 pathways (*Penuel and Martin, 1999*; ). Therefore, we evaluated the activation status of these signalling pathways in SYF cells expressing the Src variants by determining the phosphorylation levels of Erk1/2 pT202/pY204, Akt pS473, and Stat3 pY705. SYF-Src90E cells exhibited significantly elevated activation of all three pathways compared to SYFs with WT c-Src. The increase in Erk1/2 phosphorylation was similar to that observed in SYF-Src Y527F cells, while the increase in phosphorylation of Akt and Stat3 was slightly less prominent. Conversely, SYF-Src90F fibroblasts showed a trend towards reduced signalling through Ras/MAPK, PI3K/Akt, and Stat3 compared to SYF-c-Src cells, although the difference was not statistically significant (*Figure 5D*).

Taken together, these results suggest that Y90 phosphorylation itself represents a positive regulatory mechanism of Src, leading to elevated activation of mitogenic pathways and oncogenic transformation.

## Phosphomimicking Src90E variant increases cell invasiveness

Src kinase is essential for cellular movement and regulates numerous attributes of cell motility (*Guarino, 2010*). Therefore, we analysed cell invasion capability of SYF cells expressing individual Src variants. We used two types of experimental setups. For analysis of individual cell invasion, we employed a vertical 3D collagen invasion assay with cells seeded on the surface of a collagen layer and invading into the matrix. We further performed a spheroid invasion assay, which better mimics invasion from tumours with respect to growth kinetics, microenvironment (gradients of oxygen, nutrients, metabolites), and appropriate morphological and functional features of cells (*Friedrich et al., 2007*).

In both experimental setups, SYF fibroblasts expressing Src with the phosphomimicking mutation Y90E exhibited an increased ability to invade into collagen compared to SYFs with WT c-Src, albeit lower than SYF cells bearing constitutively active Src527F. SYF fibroblasts expressing the nonphosphorylatable variant Src90F invaded comparably or less, though not significantly, than SYF-c-Src cells. SYFs without Src (MSCV) were non-invasive or displayed very low invasiveness (*Figure 6A–C*).

Since integrin-mediated Src signalling was shown to have an important role in Src-promoted cellular invasiveness (*Brábek et al., 2004*; *Guarino, 2010*; *Thomas and Brugge, 1997*), we also analysed the phosphorylation status of several FA proteins which are known Src substrates involved in integrin signalling. Compared to cells with WT c-Src, we detected increased levels of phosphorylated FAK, p130Cas, and paxillin in SYF-Src527F fibroblasts. Cells expressing the phosphomimicking variant Src90E exhibited elevated phosphorylation of paxillin, yet less pronounced than SYF-Src527F cells. Phosphorylation of p130Cas was also slightly increased, although not significantly. Expression of Src90E did not affect FAK autophosphorylation on Y397, but, interestingly, reduced phosphorylation of FAK on Y861, which is a direct Src phosphorylation site. SYF fibroblasts with the Src90F mutant displayed decreased phosphorylation of FAK, p130Cas, and paxillin in comparison with SYF-c-Src cells (with statistical significance only for FAK Y861) (*Figure 6D*).

Abrogating Y90 phosphorylation dynamics by the phosphomimicking substitution 90E resulted in substantially elevated invasive potential of cells, revealing a possible role of Y90 phosphorylation in the regulation of cellular motility. However, despite its high catalytic activity and capability to enhance cell invasiveness, Src90E-mediated phosphorylation of the proteins associated with FAs and integrin signalling is mild or even lower than that of c-Src.

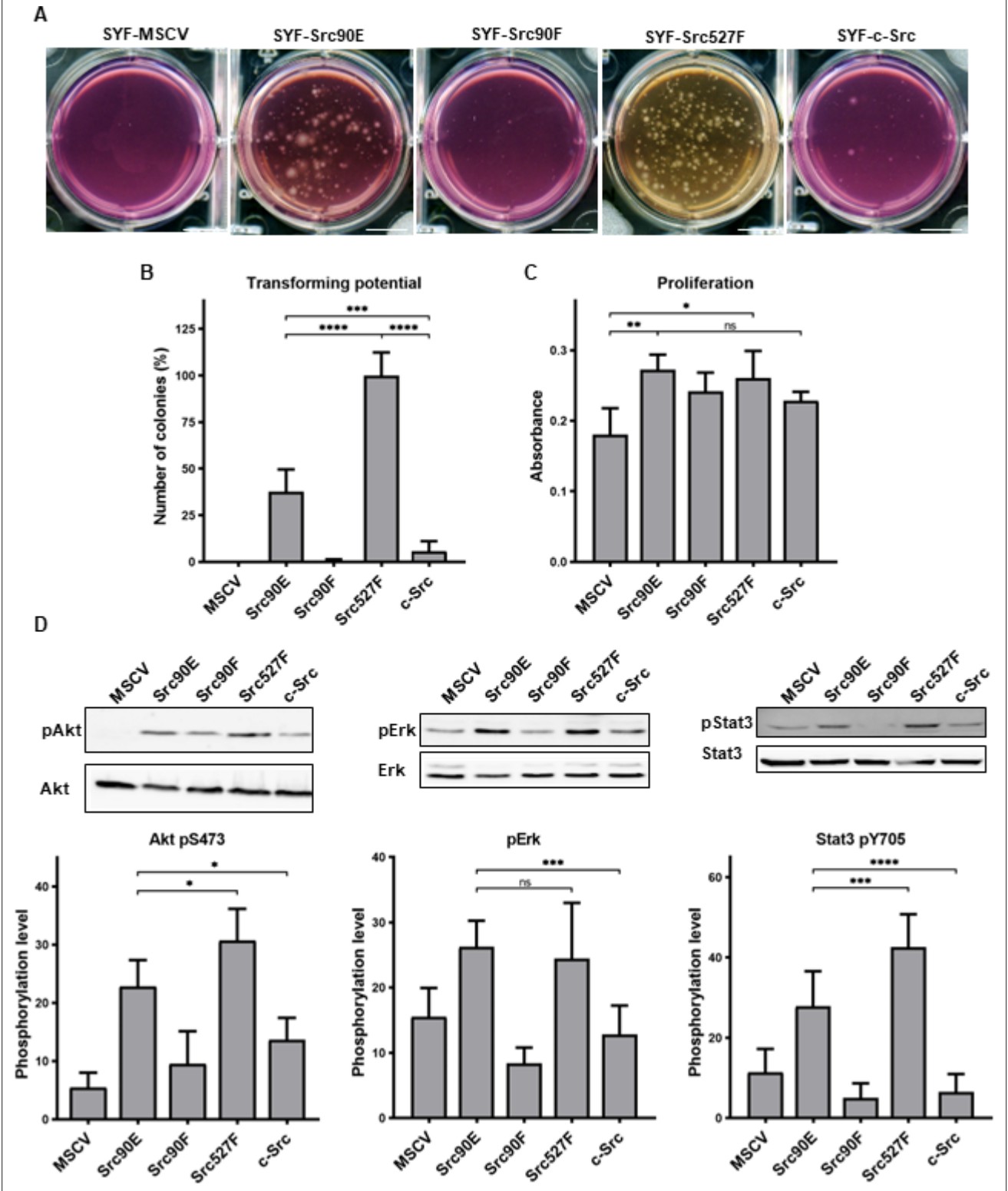

**Figure 5.** Src90E has transforming potential. (**A, B**) Soft agar assay was performed with SYF fibroblasts stably expressing the Src variants. Cells were cultivated in 0.4% agar. After 14 days the number of colonies was determined. Scale bars: 10 mm. The graph shows relative amounts of colonies compared to SYF-Src527F. (**C**) Proliferation rate determined by alamarBlue assay. Equal number of cells was seeded into 98-well plate in tetraplicates for every lineage. After 3 days, change of absorbance at 570 nm (referenced to 600 nm) of added alamarBlue solution was measured. (**D**) Activity within PI3K/Akt, Ras/MAPK, and Stat3 pathways evaluated as the phosphorylation rate of Akt, Erk, and Stat3. Lysates from SYF fibroblasts stably expressing

*Figure 5 continued on next page*

*Figure 5 continued*

the Src variants were standardised to overall protein. Immunodetection on western blots was performed with antibodies against Akt, Akt pS473, Erk 1/2, pErk 1/2, Stat3, and Stat3 pY705. Rates between phosphorylated form and total protein were calculated from at least three independent experiments and plotted as means with standard deviation. Statistical significance was determined by one-way ANOVA.

The online version of this article includes the following source data for figure 5:

**Source data 1.** Unedited immunoblots and source datasets for graphs in *Figure 5*.

## Effect of 527F-activated Src can be modulated by Y90 phosphorylation

The mutation mimicking the phosphorylation of Y90 increased Src kinase activity and its transforming and invasion-promoting potential, while the nonphosphorylatable 90F variant exhibited opposite effects. Since Y90 phosphorylation is expected to interfere with the intramolecular lock between the SH3 domain and the CD linker, we wondered whether the phosphorylation status of Y90 would be able to affect already activated Src via 527F substitution, which abrogates the binding of the SH2 domain to the C-terminus. We therefore created SYF cell lines expressing double-mutated Src variants (Src527F90E, Src527F90F) and analysed these for kinase activity. We measured a statistically significant increase in catalytic activity for the Src527F90E variant compared to Src527F and a small, though not significant, decrease of activity for Src527F90F (*Figure 7A and B*).

SH2-activated Src (Src527F) is strongly accumulated in FAs. Using double-mutated SrcFRET sensors, we tested whether substitutions of Y90 altering binding properties of the SH3 domain affect this phenotype. Both SrcFRET527F90E and SrcFRET527F90F displayed similar localisation to FAs to SrcFRET527F (*Figure 7—figure supplement 1*). We further employed these FRET-based sensor molecules to analyse structural compactness of the double-mutated variants. In the context of SH2-activated Src, the 90E mutation (SrcFRET527F90E) did not cause any additional opening of the kinase conformation, whereas the 90F substitution (SrcFRET527F90F) led to a more condensed structure in comparison with SrcFRET527F (*Figure 7C and D*).

Next, we sought to assess the effects of the double-mutated Src variants on cellular physiology. To test their transforming potential, we performed the soft agar assay. We observed a significant raise in the number of colonies while expressing Src527F90E in comparison with Src527F, indicating its higher capability to induce cellular transformation (*Figure 7E and F*). Analysing invasiveness of the cells by spheroid invasion assay revealed substantial differences, with Src527F90E enhancing and Src527F90F reducing the impact of Src527F (*Figure 7G and H*).

These results suggest that the SH2-activated Src molecule might be further modulated by the SH3 domain. We propose that the engagement of the SH3 domain in an intramolecular inhibitory lock controlled by Y90 phosphorylation represents an independent regulatory mechanism of the Src kinase affecting its catalytic activity, interactions, and cellular responses with high degree of autonomy from the SH2 domain.

## Discussion

The activity of the Src kinase is regulated mainly by the phosphorylation status of Y416 and Y527 and associated intramolecular inhibitory interaction mediated by SH3 and SH2 domains (*Boggon and Eck, 2004*). Additional regulatory levels predominantly involve phosphorylation of serines or threonines within the UD and Src dimerisation (*Amata et al., 2014*; *Dandoulaki et al., 2018*; *Spassov et al., 2018*).

In this study, we are documenting the phosphorylation of Y90 on the binding surface of the SH3 domain as a novel regulatory mechanism of the Src kinase. The phosphorylation state of Y90 controls the Src kinase on several levels. Mimicking phosphorylated Y90 by the 90E substitution revealed that phosphorylation of Y90 results in increased Src kinase activity, reduced ability of the SH3 domain to bind ligands, slower mobility within cytoplasmic membrane, and induces a more open conformation of the kinase. Expression of the phosphomimicking mutant in cells leads to cellular transformation and elevated invasiveness.

Biological relevance of the proposed Src kinase regulation by phosphorylation of Y90 is supported by the presence of homologous tyrosines in SH3 domains of various different proteins. The amino acid motif ALYDY/F flanking Y90 is conserved among SH3 domains on sequential and structural level. This

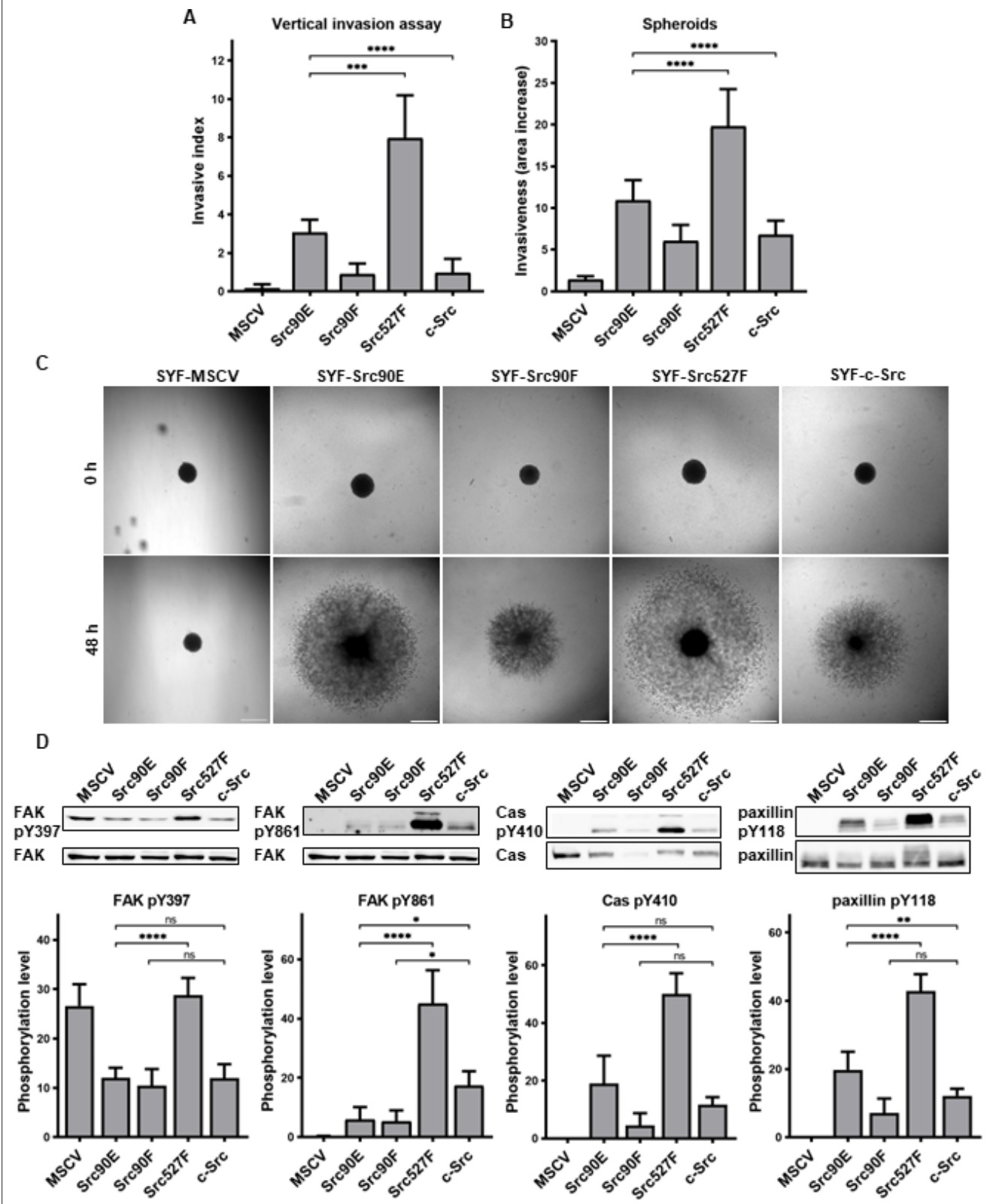

**Figure 6.** Phosphomimicking mutation 90E increases cell invasiveness. (**A**) Vertical invasion assay performed with SYF fibroblasts stably expressing the Src variants. Cells were seeded on top of 1.5% collagen. After 3 days, z-stack pictures were taken, and number of cells was counted in each level. Cellular invasiveness was determined as invasive index representing weighted arithmetic mean of invasion depth. (**B, C**) Spheroid invasion assay. SYFs grown as spheroids were embedded in 1.5% collagen. Images were taken immediately after seeding and 48 hr later. Scale bars: 500 µm. Ratio between

*Figure 6 continued on next page*

*Figure 6 continued*

spheroid area at 48 hr and 0 hr was determined. (**D**) Phosphorylation levels of FAK, p130Cas, and paxillin were detected in lysates from SYF cells stably expressing the Src variants. Ratio between phosphorylated form and total protein was determined based on immunodetection of total FAK, p130Cas, paxillin, and their phosphorylated forms (FAK pY397, FAK pY861, p130Cas pY410, paxillin pY118). All the data in bar graphs are shown as means with standard deviation out of minimum three independent experiments. Statistical significance was calculated by one-way ANOVA.

The online version of this article includes the following source data for figure 6:

**Source data 1.** Unedited immunoblots and source datasets for graphs in *Figure 6*.

sequence was found in 15% of human SH3 domains, several of which were reported to be phosphorylated (*Tatárová et al., 2012*). For example, in SH3 domains of PST-PIP, the Btk kinase or the adaptor protein p130Cas, phosphorylation of tyrosines corresponding to Y90 causes decreased or changed affinity for ligands (*Gemperle et al., 2017*; *Janoštiak et al., 2011*; *Morrogh et al., 1999*; *Park et al., 1996*; *Wu et al., 1998*). In chronic myeloid leukaemia cells, the SH3-SH2 segment of Bcr-Abl kinase was found to be heavily phosphorylated on seven residues including the Y90 homolog, which induces full catalytic activity of the fusion protein (*Chen et al., 2008*; *Meyn et al., 2006*). Src phosphorylated on Y90 was found in MEFs transformed by oncogenic Src or in non-small cell lung carcinoma cells stimulated with HGF (*Johnson et al., 2013*; *Luo et al., 2008*). Using an immunoprecipitated Src and targeted MS approach, we were able to detect and quantify the level of Y90 phosphorylation even in resting untransformed cells efficiently and consistently. Y90 is phosphorylated in approximately 1% of WT Src molecules, but in the case of activated kinase its phosphorylation increases fivefold (5.3% of Src molecules). However, compared to activation loop phosphorylation (Y416), phosphorylation of Y90 is 22-fold and 10-fold less abundant when compared to WT and activated kinase, respectively. Importantly, although the enrichment of Y90 phosphorylation in the catalytically active kinase is lower compared to Y416 phosphorylation in terms of percentage of phosphorylated molecules, its increment with respect to the basal state is significantly higher. We suggest that this wider dynamic range of Y90 phosphorylation is consistent with and reflects the demonstrated regulatory function of Y90 phosphorylation.

It has been shown that autophosphorylation of Y416 is dependent on Src dimerisation and occurs via transphosphorylation (*Spassov et al., 2018*). In our experimental setup, quantification of Y416 phosphorylation by MS was performed in cells expressing both the analysed Src biosensor and endogenous Src. Interestingly, the level of Y416 phosphorylation was negligible in the kinase-dead variant of the biosensor. This could indicate that the Src biosensor cannot form a stable dimer with endogenous Src molecules. However, since Y416 is efficiently phosphorylated in all kinase-competent biosensor variants, dimerisation of these forms of the biosensor is likely unaffected. This observation is consistent with the previously described dependence of Src dimer formation on Src kinase activity, as Src inhibitors have been shown to block de novo Src dimer formation (*Spassov et al., 2018*). We therefore propose that only if both Src molecules are kinase competent, they can form a dimer.

In the context of Src structure, Y90, together with asparagine 135 and tyrosine 136, forms the first hydrophobic pocket of the SH3 domain binding surface, which in the inactive conformation of the Src kinase accommodates a dipeptide containing the key proline 250 of the SH3-interacting motif within the CD linker (*Xu et al., 1997*). The importance of Y90 for ligand binding was mentioned in an early study mapping conserved residues within the SH3 domain (*Erpel et al., 1995*). Substitution of Y90 for alanine led to a partially deregulated Src and a reduced affinity of the SH3 domain. Based on NMR analyses and molecular dynamics simulations of ligand binding by the p130Cas SH3 domain containing a homologous tyrosine (*Gemperle et al., 2017*), we propose that the aromatic ring of unphosphorylated Y90 forms CH/π interactions with the dipeptide in the binding pocket. Through the negative charge of the phosphate group, phosphorylation of Y90 causes a disruption of these nonpolar bonds with a consequent decrease in affinity of the SH3 domain for its ligands. Y90 phosphorylation state might therefore influence the Src kinase via a dual mechanism of action: regulating its catalytic activity by modulating SH3 domain binding to the CD linker and affecting its ability to associate with interacting partners.

Molecular dynamics simulations show that residues of the CD linker are directly responsible for retaining the inactive conformation of the catalytic domain. The SH3 domain together with the SH2 domain stabilises this inhibitory position of the CD linker (*Fajer et al., 2017*). We propose that phosphorylation of Y90 within the binding surface of the SH3 domain leads to a decreased affinity for the

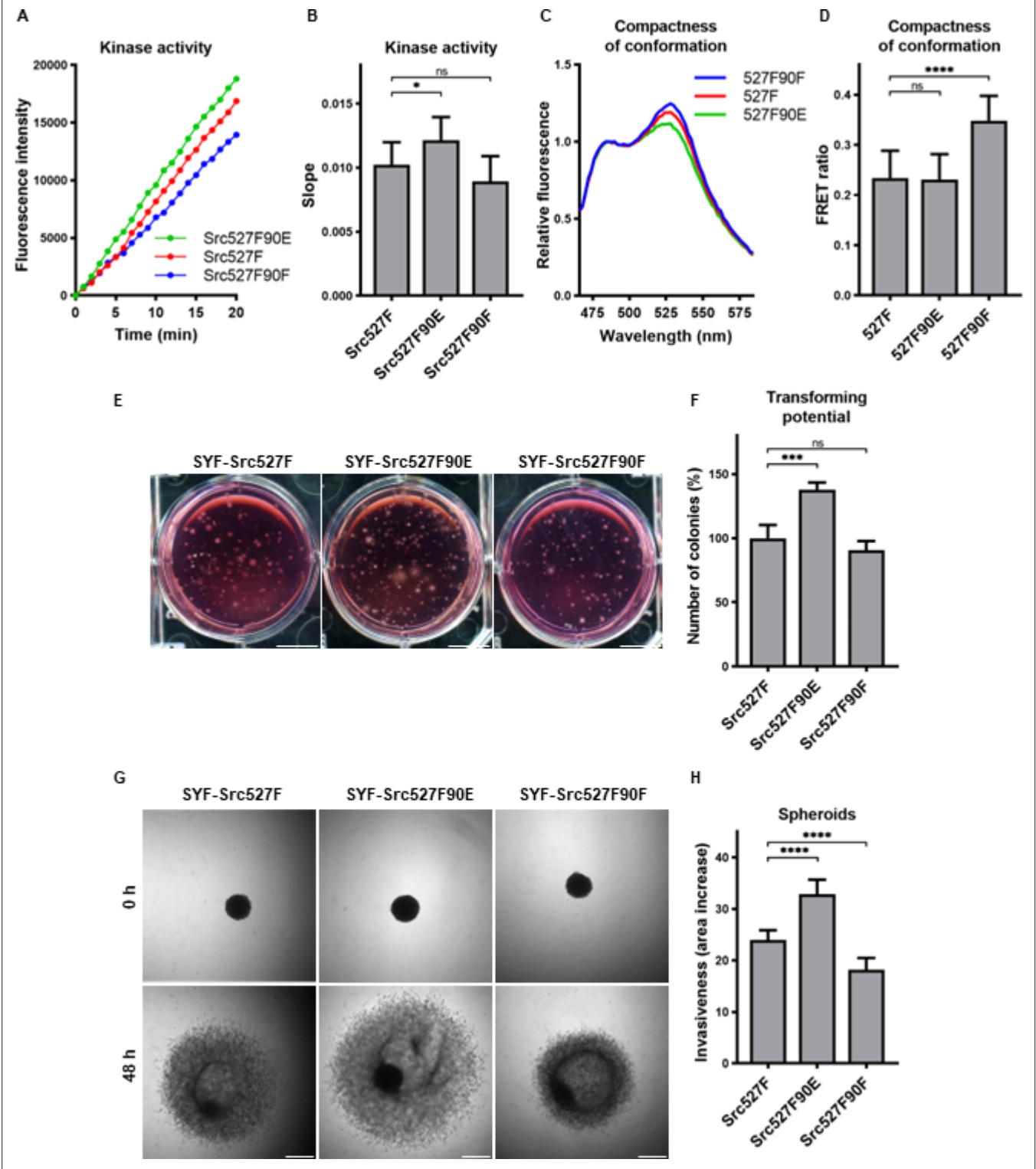

**Figure 7.** Effect of 527F-activated Src can be modulated by Y90 phosphorylation. (**A, B**) Catalytic activity of Src527F and the double-mutated variants was analysed using kinase assay (Omnia Y Peptide 2 Kit). Reactions were performed from lysates of SYF cells expressing indicated Src variants. Kinase activity was measured as an increase of fluorescence depicted in *Figure 6B* (showing representative results from one experiment). (**C, D**) Effect of Y90 substitutions on compactness of the Src527F structure. SrcFRET527F sensor variants with Y90 mutations were expressed in U2OS cells. Fluorescence emission spectra were recorded in native cell lysates. (**C**) Representative emission spectra normalised to emission maximum of CFP. PC indicates positive control for FRET. (**D**) Bar graph shows ratio of normalised mCit (525 nm) and CFP (486 nm) emission. (**E, F**) Transforming potential was assessed by

*Figure 7 continued on next page*

*Figure 7 continued*

soft agar assay with SYF fibroblasts stably expressing indicated Src variants. Cells were cultivated in 0.4% agar. After 14 days, the number of colonies was counted. Scale bars: 10 mm. The graph shows relative number of colonies compared to SYF-Src527F. (**G, H**) Invasiveness of the SYF cell lines was evaluated with spheroid invasion assay. SYFs grown as spheroids were embedded in 1.5% collagen. Images were taken immediately after seeding and 48 hr later. Scale bars: 500 µm. Ratio between spheroid area at 48 hr and 0 hr was determined. The data in bar graphs are shown as means with standard deviation out of minimum three independent experiments. Statistical significance was calculated by one-way ANOVA.

The online version of this article includes the following source data and figure supplement(s) for figure 7:

**Source data 1.** Source datasets for graphs in *Figure 7*.

**Figure supplement 1.** Localisation of double-mutated SrcFRET constructs.

CD linker, presumably followed by a destabilisation of contacts with the kinase domain and a shift to the catalytically active state.

The phosphomimicking mutant Src90E adopts a partially open conformation with the SH3 domain released from inhibitory interactions. Interestingly, the SH2 domain of Src90E appears to be still bound to the C-terminal tail. The presence of an intact SH2-C-terminus interaction is further supported by a uniform cellular distribution of Src90E, since disengaged SH2 domain causes the accumulation of Src in FAs (*Wu et al., 2015*). We therefore suggest that the Src kinase may exist not just in the SH2-activated form but also in an SH3-only activated state.

The introduction of 90E or 90F mutations into the SH2-activated kinase (via 527F substitution) affected its activity and, more importantly, its transforming potential and cellular invasiveness. Specifically, the 90E mutation emphasised, and the 90F substitution reduced, the impact of 527F-activated Src. Thus, we assume that both regulatory domains cooperate and influence the kinase structure and activity with some degree of independency, enabling Src to adopt several active conformations (SH2-activated, SH3-activated, SH2+SH3-activated). This structural hierarchy increases the functional diversity and regulatory potential of the Src kinase and its downstream signalling.

Together with the SH4 and UD, the SH3 domain ensures the ability of Src to associate with the cytoplasmic membrane and provides a structural support to this intrinsically disordered N-terminal region of Src. Membrane contacts of the SH3 domain are maintained by RT and nSrc loops, which are also responsible for binding the unique and SH4 domains, respectively (*Maffei et al., 2015*; *Pérez et al., 2013*). These SH3-mediated interactions were described to be governed by the ligand-binding state of the domain in a mutually exclusive manner where the presence of a ligand weakens the contacts of SH3 with the N-terminal domains and lipids (*Cordier et al., 2000*; *Maffei et al., 2015*; *Wang et al., 2001*). Controlling the phosphorylation state of Y90 within the SH3 domain provides a mechanism for regulation of these interactions and therefore of Src dynamics within membranes. We propose that phosphorylation of Y90 decreases the affinity of the SH3 domain towards ligands and consequently increases the association of the SH3 domain with the UD and the lipid layer, which results in slower mobility within membranes and prolonged residence in FA sites. This seems to be in agreement with the work of *Machiyama et al., 2015*, who were analysing movement of single Src molecules within the plasma membrane. They demonstrated decreased membrane motility and increased dissociation times from FAs of Src with W118A mutation, which abrogates ligand binding to the SH3 domain (*Erpel et al., 1995*).

Besides maintaining interactions with lipids and N-terminal unstructured domains, the RT and nSrc loops of the SH3 domain also form inhibitory contacts with the N-terminal lobe of the kinase domain (*Brábek et al., 2002*; *Xu et al., 1997*). Residues of the RT loop responsible for interaction with the catalytic domain overlap with some of those associating with membranes and the UD (R95, T96). We speculate that phosphorylation of Y90 will block binding of the CD linker by the SH3 domain, which might cause a shift towards a state where the SH3 domain favours interactions with the cytoplasmic membrane and the UD over keeping contacts with the N-lobe, resulting in kinase activation. The SH3 domain could be therefore perceived as a crucial element interconnecting the lipid layer, unstructured, and structured regions of Src, thus forming a signalling hub which mediates transitions between catalytically inactive and active states of the kinase. Phosphorylation of Y90 might then represent an important molecular switch facilitating these processes, hence regulating Src catalytic activity, interactions, and mobility within cell membranes.

To conclude, we present a new model for the regulation of Src by tyrosine phosphorylation. In this model, the SH2 and the SH3 domains serve as cooperative but independent regulatory elements of

the Src kinase. Their function and their engagement in the intramolecular inhibitory lock are controlled by phosphorylation on tyrosines Y527 and Y90. Both phosphorylations affect the opening of Src structure and its catalytic activity, but in opposite ways. Furthermore, unlike Y527, phosphorylation of Y90 greatly influences the repertoire of Src binding partners and enhances Src affinity for the plasma membrane. Taken together, the modular system of regulations through phosphorylation of key tyrosines and intramolecular interactions enables the Src kinase to adopt several different conformations of varying kinase activities and interacting properties. That allows Src to operate not as a simple on/off switch but as a tunable regulator functioning as a signalling hub in a variety of cellular processes.

## Materials and methods

### Plasmid constructs and cloning

To prepare SrcY90E and SrcY90F, c-Src cloned in pBluescript SK+ vector through *BamHI* and *XbaI* sites was mutated using QuikChange II Site-Directed Mutagenesis Kit (Agilent Technologies) with the forward primer (5'-CCACTTTCGTGGCTCTCGAGGACTACGAGTCCCGGACTG-3') and the reverse primer (5'-CAGTCCGGGACTCGTAGTCCTCGAGAGCCACGAAAGTGG-3') for Y90E mutation and the forward (5'-CCACTTTCGTGGCTCTCTTCGACTACGAGTCCCGGACTG-3') and the reverse (5'-CAGTCCGGGACTCGTAGTCGAAGAGAGCCACGAAAGTGG-3') primers for Y90F variant. Double-mutated Src527F90E and Src527F90F were created from pBluescript-Src90E or pBluescript-Src90F by swapping the C-terminal half of Src for the same part but from our Src527F construct using BamHI and MluI sites. After verification by sequencing, Src variants (together with SrcY527F) were recloned into pMSCV-GFP murine retrovirus vector (pMSCVpuro vector [Clontech] where puromycin resistance gene was exchanged for *gfp*; a kind gift from Dr. Michal Dvořák [Institute of Molecular Genetics of the ASCR, Prague, Czech Republic]). Src fragments were generated using *NotI* (blunt ended by *Pfu* polymerase) and *BamHI* restriction endonucleases from pBluescript constructs and inserted to *HpaI* and *BglII* sites of pMSCV-GFP vector. GST-fusion constructs of SH3 domains (WT, v-Src, Y90E, Y90F) for pull-down experiments were prepared by PCR amplification from the corresponding pBluescript plasmids with the forward primer (5'-GGATCCATGGCTGGCGGCGTCACC-3') incorporating a *BamHI* site prior to the start codon and the reverse primer (5'-GAATTCTAGATGGAGTCTGAGGGCGCG-3') adding an *EcoRI* site and the stop codon. The cleaved products were inserted into pGEX-2T vector (GE Healthcare Life Sciences) via *BamHI* and *EcoRI* sites resulting in SH3 domain constructs carrying the GST tag at their N-termini. SrcFRET sensors bearing 90E or 90F mutation were prepared form SrcFRET (SrcFRET527F) (*Koudelková et al., 2019*) and Src90E (Src90F) constructs in pBlueScript SK+ vector by the swapping 5'-end part of the gene using BamHI/BsmBI restriction sites. The resulted constructs were cloned into BamHI/NotI sites of expression vector pIRESpuro3.

### Cell culture

HeLa cells, Phoenix cells, U2OS (ATCC, cat. n. HTB-96), and SYF (*Src−/−, Yes−/−, Fyn−/−*) MEFs (*Klinghoffer et al., 1999*) (ATCC, cat. n. CRL-2459) were cultured at 37°C with 5% $CO_2$ in DMEM (Sigma) with 4.5 g/l L-glucose, L-glutamine, and pyruvate, supplemented with 10% fetal bovine serum (Sigma) and 50 µg/ml Gentamicin (Sigma). The source of all parental lines used was ATCC. Cells were mycoplasma free (tested by PCR). Cell transfections were carried out using Jet Prime (Polyplus Transfection) according to the manufacturer's protocol. Stable cell lines of SYF cells expressing Src variants were prepared using MSCV constructs, which were transfected into the Phoenix retroviral packaging lineage. SYF fibroblasts were infected with obtained retroviral particles and subsequently sorted by FACS for GFP-positive cells.

### Immunoblotting

Cells were washed with phosphate-buffered saline (PBS) and lysed in modified RIPA buffer (0.15 M NaCl; 50 mM Tris-HCl, pH 7.4; 1% Nonidet P-40; 0.1% SDS; 1% sodium deoxycholate; 5 mM EDTA; 50 mM NaF; 1 mM dithiothreitol; protease and phosphatase inhibitor cocktail [Sigma]). Lysates were clarified by centrifugation and protein concentration was determined using DC Protein Assay (Bio-Rad). Samples were supplemented with Laemmli buffer (0.35 M Tris-HCl, pH 6.8; 10% SDS; 40% glycerol; 0.012% bromophenol blue, 50 mM dithiothreitol), boiled for 5 min, separated by SDS-PAGE electrophoresis and transferred to nitrocellulose membranes. Membranes were then blocked

with Tris-buffered saline containing 4% bovine serum albumin and incubated with primary and HRP-conjugated secondary antibodies. Blots were developed using Fuji LAS-1000 chemiluminescence imaging system. Western blot quantification was carried out using the ImageJ software (https://imagej.nih.gov/ij/).

## Antibodies

Antibodies for western blot detection included actin (I-19, Santa Cruz Biotechnology, cat. n. sc-1616, RRID: AB_630836), Akt (Cell Signaling Technology, cat. n. 9272, RRID: AB_329827), Akt pS473 (Cell Signaling Technology, cat. n. 9271, RRID: AB_329825), Cas (BD Transduction Laboratories, cat. n. 610271, RRID: AB_397666), Cas pY410 (Cell Signaling Technology, cat. n. 4011, RRID: AB_2274823), Erk1/2 (Promega), phospho-Erk1/2 (Promega, cat. n. V1141, RRID: AB_430839), FAK (C-20, Santa Cruz Biotechnology, cat. n. sc-558, RRID: AB_2300502), FAK pY397 (Invitrogen, cat. n. 44–624G, RRID: AB_2533701), FAK pY861 (Abgent, cat. n. AJ1285f, RRID: AB_10818227), paxillin (BD Transduction Laboratories, cat. n. 610052, RRID: AB_397464), paxillin pY118 (Cell Signaling Technology, cat. n. 2541, RRID: AB_2174466), Src (clone 184Q20, Invitrogen, cat. n. AHO1152, RRID: AB_2536324), Src pY418 (Cell Signaling Technology, cat. n. 2101, RRID: AB_331697), Stat3 (C-20, Santa Cruz Biotechnology, cat. n. sc-482, RRID: AB_632440), and Stat3 pY705 (Cell Signaling Technology, cat. n. 9131, RRID: AB_331568).

## Kinase assay

Kinase activity of Src variants was measured using Omnia Y Peptide 2 Kit (Thermo Fisher Scientific). The assay is based on the detection of fluorescence increase after kinase-specific substrate phosphorylation. Specifically, a substrate peptide is attached to the chelation-enhanced fluorophore Sox. Upon phosphorylation of the peptide by the kinase, $Mg^{2+}$ is chelated to form a bridge between the Sox moiety and the incorporated phosphate group on the tyrosine within the substrate peptide, which consequently causes increase in fluorescence. Kinase assays were performed according to the manufacturer's protocol. Briefly, cells expressing equivalent amounts of Src variants were washed with PBS and lysed in HEPES-Triton buffer (1% Triton X-100, 50 mM HEPES pH 7.4, protease and phosphatase inhibitors). Lysates were clarified by centrifugation and diluted to reach identical protein concentrations. Kinase reactions (in triplicates for each variant) were assembled by adding cell lysate, kinase reaction buffer, 0.2 mM DTT, 1 mM ATP, and 10 µM peptide substrate. Using a plate reader (TECAN Infinite M200 PRO), reactions were incubated at 30°C and fluorescence intensity was measured ($\lambda_{ex}$ 360/$\lambda_{em}$ 485) at 60 s intervals for 1 hr.

## Purification of GST-fused SH3 domains and GST pull-down assay

Bacterial cells BL21 were transformed with pGEX-SH3 plasmid constructs. While cultivating, expression of GST-fused variants of SH3 domains was induced by adding IPTG (1 mM concentration). After 2 hr bacteria were centrifuged, washed in LB1 buffer (50 mM HEPES, pH 7.4; 100 mM NaCl), and lysed by sonication. Clarified bacterial lysates were incubated with glutathione sepharose beads 4B (GE Healthcare Life Sciences). Beads with immobilised GST-SH3 domains were extensively washed in LB1 buffer and then added to lysates from HeLa cells. Two hr of incubation was followed by washing of beads in LB1 buffer. Precipitated proteins were eluted by boiling the beads in Laemmli sample buffer and detected using SDS-PAGE and immunoblotting.

## Immunoprecipitation

SYF cells expressing Src variants were lysed in Tris-Triton buffer (1% Triton X-100, 50 mM Tris HCl pH 7.5, 150 mM NaCl, 1 mM DTT, protease and phosphatase inhibitors). Lysates were clarified by centrifugation and protein concentration was determined using DC Protein Assay (Bio-Rad). Lysates of equal protein concentration were incubated overnight in 4°C with an anti-Src antibody (Cell Signaling Technology, cat. n. 2108, RRID: AB_331137). Complexes were pulled out using protein-A sepharose beads (GE Healthcare), washed four times with Triton-Tris buffer, and eluted by boiling for 10 min in 2× Laemmli buffer.

## Soft agar assay

A 1.6% solution of melted agar (Noble agar, Sigma) was mixed with warm 2× DMEM with 20% FBS and antibiotics. This 0.8% base agar was used to coat a 35 mm dish (six-well plate). One ml of 0.8%

melted base agar was further diluted with equal volume of warm (40°C) 1× DMEM (with 10% FBS and antibiotics) containing $2×10^3$ cells and added on top of the solidified layer of base agar. After stabilisation of the top layer, agar was covered with medium and kept in an incubator for 14 days. The number of colonies was then counted in each dish.

## Proliferation assay

$5×10^3$ cells (in quadruplicates per cell line) were plated on a 96-well plate and grown for 3 days. Proliferation was then analysed using alamarBlue (Life Technologies) according to the manufacturer's protocol. Briefly, alamarBlue reagent was added as 10% of the sample volume to cells and after 2 hr of incubation absorbance at 570 nm (normalised to 600 nm as reference wavelength) was read on a spectrophotometer (TECAN Infinite M200 PRO).

## Vertical invasion assay

Cell invasiveness was analysed in 1.5 mg/ml collagen gel (rat tail, collagen type I). Collagen mixture (1.5 mg/ml rat-tail collagen, 1× DMEM, 1% FBS) was added into each well of a µ-Slide Angiogenesis plate (Ibidi) and polymerised at 37°C. $3×10^3$ cells in DMEM (containing 10% FBS and antibiotics) were seeded on top of the collagen gel and allowed to attach. Next day medium was aspirated and replaced with DMEM with 1% FBS. After 3 days, z-stack pictures were taken on Nikon Eclipse TE2000-S microscope (20×/0.40 HMC objective; NIS-Elements software). Number of cells in each layer of a z-stack was determined using ImageJ. Invasiveness was calculated as an arithmetic mean of cell count weighted by invasion depth in the selected field. For each cell line, invasion was analysed in three wells (nine fields of view) per one independent experiment.

## Spheroid invasion assay

SYF cells were grown as spheroids using 3D Petri Dish (Microtissues) according to the manufacturer's instructions. After 2 days of spheroid formation, 1.5 mg/ml collagen solution (rat tail, collagen type I) was prepared (1.5 mg/ml rat-tail collagen, 1× DMEM, 1% FBS) and added at the bottom of a 96-well plate. Spheroids were embedded into the collagen (one spheroid per well) and covered with another layer of the gel. Collagen was polymerised at 37°C and subsequently overlaid with cultivation medium. Images of spheroids were taken with Nikon Eclipse TE2000-S microscope (4×/0.13 PHL objective; NIS-Elements software) immediately upon their seeding into collagen and after 48 hr. Area of spheroids at 0 and 48 hr was measured using ImageJ. Cellular invasiveness was determined as the increase of spheroid area within 48 hr. For one independent experiment, at least eight spheroids were analysed per each cell line.

## FRET measurements

U2OS cells transfected with Src sensor plasmid constructs were lysed in HEPES-Triton buffer (1% Triton X-100, 50 mM HEPES pH 7.4, 1 mM DTT, protease and phosphatase inhibitors). Lysates were clarified by centrifugation. One hundred and fifty µl of the sample (in triplicates for each biosensor variant) was transferred into a 96-well flat-bottom plate and measured using a spectrophotometer (TECAN Infinite M200 PRO). Data were collected for a series of emission wavelengths with a fixed excitation length. The series were taken in 2 nm increments starting at 466 nm and finishing at 584 nm. FRET efficiency was determined as the 528/486 nm emission ratio.

## Cell immunostaining and confocal microscopy

Cells were seeded on fibronectin-coated coverslips (10 µg/ml, Invitrogen) and grown for 24 hr. Next, cells were fixed in 4% paraformaldehyde in PBS, permeabilised using 0.3% Triton X-100 in PBS, and blocked in 3% bovine serum albumin in PBS. Samples were then incubated for 4 hr with primary antibody, 1 hr with secondary antibody, and with phalloidin for 30 min with extensive washing with PBS between each step. The slides were mounted in Mowiol 4-88 (Millipore) containing 1,4-diazobicyclo-[2.2.2]-octane (Sigma). The secondary antibody used was anti-mouse-IgG Alexa Fluor 633 (Invitrogen). F-actin was probed with phalloidin conjugated with Alexa Fluor 555 (Life Technologies). Images were acquired using Leica TCS SP8 microscope system equipped with Leica 63×/1.45 NA oil objective.

## MS analysis

U2OS cells transfected with SrcFRET constructs were lysed in HEPES-dodecylmaltoside buffer (1% $n$-dodecyl β-D-maltoside, 50 mM HEPES pH 7.4, 1 mM DTT, protease and phosphatase inhibitors). Lysates were clarified by centrifugation, equalise for concentration of SrcFRET molecules using fluorescence intensity measurements of mCitrine (part of SrcFRET constructs) and incubated overnight in 4°C with anti-GFP antibody (Life Technologies, cat. n. A-11120, RRID: AB_221568). Complexes were pulled out using protein-A sepharose beads (GE Healthcare) and washed three times with lysis buffer and twice with TBS (50 mM Tris HCl pH 7.1, 150 mM NaCl). Immunoprecipitated proteins were digested using trypsin and analysed by LC/MS. The relative amount of Src phosphorylated on Y90 was determined as normalised ratio of intensities of peptides containing phosphorylated Y90 and non-modified base peptides. Absolute quantification of peptides was done by calculating intensity ratios between peptides of interest and corresponding stable isotope-labelled standards with known concentration. Sequences of labelled standard peptides were as follows: AGALAGGVTTFVALYDYESR, AGALAGGVTTFVALY[Phospho (Y)]DYESR, LIEDNEYTAR, LIEDNEY[Phospho (Y)]TAR (JPT Peptide Technologies GmbH, Germany). Each injection contained 0.2 pmol of labelled standard peptide.

MS analysis was done on Orbitrap Ascend (Thermo Scientific). Precursors of both labelled and non-labelled peptides were isolated by quadrupole and fragmented in HCD cell using normalised collision energy of 25%. Fragments were detected in Orbitrap with 30,000 resolution. All data were further processed in Skyline-daily (version 22.2.1.425).

## Fluorescence correlation spectroscopy measurements

Mobility of Src molecules was determined using Imaging FCS. U2OS cells transfected with SrcFRET constructs and mCherry-vinculin were plated on fibronectin-coated glass-bottom dishes. Experiments were performed on Nikon Eclipse Ti-E microscope equipped with an H-TIRF module and a Nikon CFI HP Apo TIRF 100× Oil NA 1.49 objective. Time-lapse images (10,000 frames, 365 frames per second, ROI 32×32 pixels) were acquired with Andor iXon Ultra DU897 camera (Andor Technologies) using 488 nm and 561 nm excitation wavelengths for SrcFRET constructs and mCherry-vinculin, respectively. Data were analysed with the ImageJ software using the Imaging FCS plugin. Diffusion coefficients were calculated from relative coefficients obtained from ImFCS and calibration measurements of SrcFRET WT mobility with line-scan FCS approach. Scanning FCS was performed on Leica TCS SP8 microscope equipped with Leica HC PL APO CS2 63× Oil NA 1.4 objective and HydraHarp400 (Picoquant) TCSPC module. Acquired data were processed and analysed using in-house-developed software (*Benda et al., 2015*).

## Acknowledgements

The authors thank Kateřina Strouhalová and Aneta Škarková for the proofreading of the manuscript and Marie Charvátová for technical help. This research was funded by Czech Science Foundation grant 19-03932S and by the project 'National institute for cancer research' (LX22NPO5102) of the Ministry of Education, Youth, and Sports of the Czech Republic. We acknowledge the Imaging Methods Core Facility at BIOCEV, an institution supported by the MEYS CR (Large RI Project LM2018129 Czech-BioImaging) and ERDF (project No. CZ.02.1.01/0.0/0.0/18_046/0016045) for their support with obtaining imaging data presented in this paper.

The authors declare no competing financial interests.

## Additional information

### Funding

| Funder | Grant reference number | Author |
| --- | --- | --- |
| Grantová Agentura České Republiky | 19-03932S | Lenka Koudelková<br>Jan Brábek<br>Daniel Rösel |

| Funder | Grant reference number | Author |
|---|---|---|
| Ministerstvo Školství, Mládeže a Tělovýchovy | LX22NPO5102 | Lenka Koudelková<br>Daniel Rösel<br>Jan Brábek<br>Markéta Pelantová |

The funders had no role in study design, data collection and interpretation, or the decision to submit the work for publication.

## Author contributions

Lenka Koudelková, Conceptualization, Investigation, Visualization, Methodology, Writing – original draft, Writing – review and editing; Markéta Pelantová, Formal analysis, Investigation, Methodology; Zuzana Brůhová, Martin Sztacho, Vojtěch Pavlík, Dalibor Pánek, Investigation, Methodology; Jakub Gemperle, Conceptualization, Funding acquisition, Investigation, Methodology, Writing – original draft, Writing – review and editing; Pavel Talacko, Conceptualization, Formal analysis, Funding acquisition, Methodology, Writing – review and editing; Jan Brábek, Conceptualization, Formal analysis, Funding acquisition, Investigation, Methodology, Writing – review and editing; Daniel Rösel, Conceptualization, Formal analysis, Funding acquisition, Methodology, Writing – original draft, Writing – review and editing

## Author ORCIDs

Lenka Koudelková (ID) http://orcid.org/0000-0003-0635-4232
Daniel Rösel (ID) http://orcid.org/0000-0001-7221-8672

## Decision letter and Author response

Decision letter https://doi.org/10.7554/eLife.82428.sa1
Author response https://doi.org/10.7554/eLife.82428.sa2

# Additional files

## Supplementary files

• MDAR checklist

## Data availability

All data generated or analysed during this study are included in the manuscript and supplementary figures. Source Data files have been provided for Figures 1-7.

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
