## [Editor Report]

This manuscript reports that a phosphomimetic mutation of a previously unstudied phosphorylation site in the Src kinase SH3 domain (Y90) elevates Src kinase activity via loosening the conformation of the Src catalytic domain. This work establishes a solid foundation for the further analysis of how and under what circumstances this modification impacts Src kinase functions in myriad biological contexts. This paper will be of interest to those studying protein kinase domain dynamicity and structure, and also those studying the Src kinase family in cell signaling.

---

## [Decision Letter]

**Decision letter after peer review:**

Thank you for submitting your article "Phosphorylation of tyrosine 90 in SH3 domain is a new regulatory switch controlling Src kinase" for consideration by *eLife*. Your article has been reviewed by 3 peer reviewers, one of whom is a member of our Board of Reviewing Editors, and the evaluation has been overseen by Volker Dötsch as the Senior Editor. The following individual involved in the review of your submission has agreed to reveal their identity: Miquel Pons (Reviewer #3).

Essential revisions:

1) All three reviewers considered that the major limitation of the study was the lack of evidence that Y90 phosphorylation occurred to a significant extent in cells. Therefore, addressing the reviewers' suggestions and concerns relative to this point is necessary: Reviewer 1, point 3; Reviewer 2, point 2 in the public review and also Reviewer 2 recommendations; Reviewer 3's comment "The biological relevance of this mechanism remains an open question".

2) Determining whether or not Y90 phosphorylation can occur via autophosphorylation is necessary.

3) Please also address Reviewer #1 points 1, 2, and minor comments; Reviewer #2, point 1, and include missing references and discussion indicated by Reviewer #3.

*Reviewer #2 (Recommendations for the authors):*

Overall, I think the biochemical and cellular studies in this manuscript are fairly convincing and the authors' model of how Y90 phosphorylation would be expected to affect the regulation of this kinase makes intuitive sense. However, it is difficult to view these results as being biologically significant without a better idea of whether Y90 phosphorylation actually plays a regulatory role in signaling. No data on how widely this phospho-event is observed are provided and there is a lack of evidence that Y90 phospho-levels are modulated under specific signaling regimes.

*Reviewer #3 (Recommendations for the authors):*

The kinase phosphorylating Y90 may be c-Src itself (autophosphorylation) but if there are other kinases involved is an open question. Detection of pY90 in a kinase-dead Src variant may help identify other kinases.

The Y90E mutation has a clear effect on the kinase activity, probably through c-Src opening. The implied mechanism is a decrease in ligand binding by the SH3 domain that would affect not only the intramolecular interaction with the connector but also other activating intermolecular interactions. The result may be a change in the specificity of the mutated kinase. Have the authors any phosphoproteomic evidence in this direction? I am not suggesting the authors embark on a complete phosphoproteomics study, which would be beyond the scope of the present paper but it may be interesting for future studies to explore the effects of various mutations at the position of Y90.

Lipid-mediated Src dimerization involving the positively charged residues in the SH4 domain, which also interact with the SH3 domain, has been observed in constructs containing the myristoylated Src N-terminal regulatory element and full-length Src (bioRxiv 2022.05.31.494233; doi: https://doi.org/10.1101/2022.05.31.494233). Do Y90 mutations affect Src dimerization?

[Editors' note: further revisions were suggested prior to acceptance, as described below.]

Thank you for resubmitting your work entitled "Phosphorylation of tyrosine 90 in SH3 domain is a new regulatory switch controlling Src kinase" for further consideration by *eLife*. Your revised article has been evaluated by Volker Dötsch (Senior Editor) and a Reviewing Editor.

All three reviewers were impressed by the new manuscript data that quantified the extent of Y90 and Y416 phosphorylation sites and have concluded that the manuscript has been significantly improved. However, there are two remaining issues with the text that need to be addressed. First, as noted in both the first reviews and also in the review of the revised manuscript, it is important that the findings are not exaggerated. Please update the text accordingly in all relevant sections (see reviewer #2). Second, please address the concern of reviewer #3 in the updated text.

*Reviewer #2 (Recommendations for the authors):*

The major additions to the revised version of this manuscript are data quantifying the levels of Y90 phosphorylation on SrcFRET constructs using mass spectrometry with isotopic standard peptides. This is impressive work from a technical standpoint that shows that Y90 phosphorylation levels are increased in activated Src (Y527F). Unfortunately, the authors were not able to measure Y90 phosphorylation levels on endogenous Src, which somewhat diminishes the biological relevance of these results and leaves the question of whether endogenous Src is phosphorylated to any significant degree on Y90 unanswered. It also does not provide a clearer picture of how Y90 phosphorylation could play a role in Src's cellular function. Therefore, I suggest the authors further minimize their claims about the role of Y90 phosphorylation in regulating Src's cellular activity and, instead, focus more on the mechanistic and structural insight they have obtained. In addition to the new experimental data described above, the authors have provided additional experimental details that alleviate the technical concerns raised during the previous round of review.

*Reviewer #3 (Recommendations for the authors):*

The detailed quantification of the phosphphoryltion levels of Y90 and Y416 is an important addition to the paper, as well as the experiments with a kinase dead mutant to test autophosphorylation. These were key experiments requested by the reviewers that have been addressed. Overall, the paper shows a reasonably convincing view of the role of Y90 as a regulatory element of c-Src and an estimate of its relative importance with respect to other regulatory events. As a result I would recommend the publication of the paper in its present form.

Having said that, maybe the authors could comment on the lack of phosphorylation of Y416 in the kinase dead mutant even in its open 527F form in spite of the presence of endogenous Src. Src autophosphorylation is supposed to be in trans (i.e. caused by a second Src molecule) and therefore phosphorylation of Y416 in the kinase dead mutant is expected. The fact that is not observed lowers the relevance of the lack of phosphorylation of Y90. This may be related to the use of the SrcFRET variant to enhance the yield of the IP. Maybe the presence of the fluorescent proteins in the sensor has an effect on the trans autophosphoryation?

---

## [Author Response]

Essential revisions:Reviewer #2 (Recommendations for the authors):Overall, I think the biochemical and cellular studies in this manuscript are fairly convincing and the authors' model of how Y90 phosphorylation would be expected to affect the regulation of this kinase makes intuitive sense. However, it is difficult to view these results as being biologically significant without a better idea of whether Y90 phosphorylation actually plays a regulatory role in signaling. No data on how widely this phospho-event is observed are provided and there is a lack of evidence that Y90 phospho-levels are modulated under specific signaling regimes.

We repeatedly tried to obtain an antibody against phosphorylated Y90 which would allow us to assess the role of Y90 phosphorylation in Src-mediated signaling in depth, but the antibody was never convincingly specific. Thus, we validated existence of Y90 phosphorylation on MS and focused predominantly on cellular studies. We agree that data demonstrating the presence of Y90 phosphorylation in cells could be more robust. As noted above, we therefore included additional quantification of Y90 phosphorylation using MS analysis with labeled peptide standards enabling precise analysis of the amount of phosphorylated molecules in cells. The new results suggest that phosphorylation of Y90 does not represent a rare phosphorylation event but is an important regulatory element of Src activation. We believe that the combination of these approaches provides solid evidence of the regulatory role of Y90 phosphorylation in Src kinase activation, localization and cellular processes.

Reviewer #3 (Recommendations for the authors):The kinase phosphorylating Y90 may be c-Src itself (autophosphorylation) but if there are other kinases involved is an open question. Detection of pY90 in a kinase-dead Src variant may help identify other kinases.

We performed the experiment proposed by the reviewer and found that Y90 phosphorylation in SH2-activated Src (527F), which carries the kinase-dead K295M mutation, is undetectable despite the open conformation of this variant and presence of endogenous Src and other SFKs in the U2OS cells used for the experiments. These results suggest that phosphorylation of Y90 depends on catalytic activity of the kinase and is therefore very likely autophosphorylation. We added these newly obtained data to the manuscript.

The Y90E mutation has a clear effect on the kinase activity, probably through c-Src opening. The implied mechanism is a decrease in ligand binding by the SH3 domain that would affect not only the intramolecular interaction with the connector but also other activating intermolecular interactions. The result may be a change in the specificity of the mutated kinase. Have the authors any phosphoproteomic evidence in this direction? I am not suggesting the authors embark on a complete phosphoproteomics study, which would be beyond the scope of the present paper but it may be interesting for future studies to explore the effects of various mutations at the position of Y90.Lipid-mediated Src dimerization involving the positively charged residues in the SH4 domain, which also interact with the SH3 domain, has been observed in constructs containing the myristoylated Src N-terminal regulatory element and full-length Src (bioRxiv 2022.05.31.494233; doi: https://doi.org/10.1101/2022.05.31.494233). Do Y90 mutations affect Src dimerization?

We did not perform any phosphoproteomics study with our SrcY90 mutants. However, we did investigate the effect of the Y90E variant on tyrosine phosphorylation of several known Src substrates. Although we mostly observed increased phosphorylation when compared to WT Src, the phosphorylation of FAK on Y861 was significantly lower than that in case of WT Src (Figure 6D). These data were not discussed in detail in the manuscript because the literature is inconclusive as to the significance of the Src SH3 domain-dependent binding of Src to FAK. Although in suspension cells, Src binding to FAK has been shown to be independent of Src SH2 and conversely dependent on the Src SH3 domain (PMID: 11839732).

Regarding the effect of Y90 mutations on dimerization, we believe that they do indeed affect dimerization. We have some indirect evidence from line scan FCS suggesting that Y90E occurs more frequently as a dimer on the plasma membrane than other variants. However, due to the "high noise" in the data, the results were inconclusive and therefore we do not show them in the manuscript.

[Editors' note: further revisions were suggested prior to acceptance, as described below.]

All three reviewers were impressed by the new manuscript data that quantified the extent of Y90 and Y416 phosphorylation sites and have concluded that the manuscript has been significantly improved. However, there are two remaining issues with the text that need to be addressed. First, as noted in both the first reviews and also in the review of the revised manuscript, it is important that the findings are not exaggerated. Please update the text accordingly in all relevant sections (see reviewer #2). Second, please address the concern of reviewer #3 in the updated text.Reviewer #2 (Recommendations for the authors):The major additions to the revised version of this manuscript are data quantifying the levels of Y90 phosphorylation on SrcFRET constructs using mass spectrometry with isotopic standard peptides. This is impressive work from a technical standpoint that shows that Y90 phosphorylation levels are increased in activated Src (Y527F). Unfortunately, the authors were not able to measure Y90 phosphorylation levels on endogenous Src, which somewhat diminishes the biological relevance of these results and leaves the question of whether endogenous Src is phosphorylated to any significant degree on Y90 unanswered. It also does not provide a clearer picture of how Y90 phosphorylation could play a role in Src's cellular function. Therefore, I suggest the authors further minimize their claims about the role of Y90 phosphorylation in regulating Src's cellular activity and, instead, focus more on the mechanistic and structural insight they have obtained. In addition to the new experimental data described above, the authors have provided additional experimental details that alleviate the technical concerns raised during the previous round of review.

We have deleted the penultimate paragraph of the Discussion where we speculated on the potential role of Y90 phosphorylation in focal adhesion. In addition, we have removed from the last paragraph of the Discussion a sentence with a rather strong statement about the relative importance of Y90 phosphorylation.

Reviewer #3 (Recommendations for the authors):The detailed quantification of the phosphphoryltion levels of Y90 and Y416 is an important addition to the paper, as well as the experiments with a kinase dead mutant to test autophosphorylation. These were key experiments requested by the reviewers that have been addressed. Overall, the paper shows a reasonably convincing view of the role of Y90 as a regulatory element of c-Src and an estimate of its relative importance with respect to other regulatory events. As a result I would recommend the publication of the paper in its present form.Having said that, maybe the authors could comment on the lack of phosphorylation of Y416 in the kinase dead mutant even in its open 527F form in spite of the presence of endogenous Src. Src autophosphorylation is supposed to be in trans (i.e. caused by a second Src molecule) and therefore phosphorylation of Y416 in the kinase dead mutant is expected. The fact that is not observed lowers the relevance of the lack of phosphorylation of Y90. This may be related to the use of the SrcFRET variant to enhance the yield of the IP. Maybe the presence of the fluorescent proteins in the sensor has an effect on the trans autophosphoryation?

We have now included a new paragraph in the discussion that deals with this. However, we believe that our results, in the context of previously published data, strongly suggest that the biosensor is capable of dimer formation, and we conclude that two Src molecules can form a dimer only if both Src molecules are kinase competent.